# Regeneration of duckweed (*Lemna turonifera*) involves genetic molecular regulation and cyclohexane release

Lin Yang[1], Jinge Sun[1], Congyu Yan[1], Junyi Wu[1], Yaya Wang[1], Qiuting Ren[1], Shen Wang[1], Xu Ma[1], Ling Zhao[2], Jinsheng Sun[1]*

1 Tianjin Key Laboratory of Animal and Plant Resistance, College of Life Sciences, Tianjin Normal University, Tianjin, China, 2 Department of Plant Biology and Ecology, College of Life Sciences, Nankai University, Tianjin, China

* skysjs@tjnu.edu.cn

**Data Availability Statement:** All relevant data are within the manuscript and its Supporting Information files.

## Abstract

Plant regeneration is important for vegetative propagation, detoxification and the obtain of transgenic plant. We found that duckweed regeneration could be enhanced by regenerating callus. However, very little is known about the molecular mechanism and the release of volatile organic compounds (VOCs). To gain a global view of genes differently expression profiles in callus and regenerating callus, genetic transcript regulation has been studied. Auxin related genes have been significantly down-regulated in regenerating callus. Cytokinin signal pathway genes have been up-regulated in regenerating callus. This result suggests the modify of auxin and cytokinin balance determines the regenerating callus. Volatile organic compounds release has been analysised by gas chromatography/ mass spectrum during the stage of plant regeneration, and 11 kinds of unique volatile organic compounds in the regenerating callus were increased. Cyclohexane treatment enhanced duckweed regeneration by initiating root. Moreover, Auxin signal pathway genes were down-regulated in callus treated by cyclohexane. All together, these results indicated that cyclohexane released by regenerating callus promoted duckweed regeneration. Our results provide novel mechanistic insights into how regenerating callus promotes regeneration.

## Introduction

Regeneration of entire plants from callus *in vitro* depends on pluripotent cell mass, which provides generates a new organ or even an entire plant [1, 2]. Regeneration was widely used for vegetative propagation of excellent variety, detoxification and obtaining transgenic crops [3, 4]. Numerous studies have focused on the molecular framework of de novo organ formation in *Arabidopsis thaliana*. The molecular factors of cellular pluripotency during the regeneration of plants have been thoroughly investigated. However, the regulatory modules in monocot plants have not been studied in-depth. Duckweed, with the advantages of fast reproduction, high protein content [5], and tolerance for various toxic substances [6, 7], has been applied as

**Funding:** Funded studies Initials of the authors who received each award: L.Yang., JG. Sun Grant numbers awarded to each author: 86185 dollars, 9276 dollars and 785 dollars The full name of each funder: Present research has been supported by National Natural Science Foundation of China (No. 32071620), Tianjin Natural Science Foundation of Tianjin (S20QNK618), The Postgraduate Innovative Research Projects of Tianjin (2020YJSS133) URL of each funder website: N/A Did the sponsors or funders play any role in the study design, data collection and analysis, decision to publish, or preparation of the manuscript? Yes.

**Competing interests:** NO authors have competing interests. There is no interest conflict in submitting this paper, and all authors have approved this manuscript for publication. This work is an original research that has not been published previously and is not considered for other publications elsewhere. All of us listed have approved this manuscript.

a monocotylous model plant for gene-expression systems. In duckweed, stable transformation mediated by *Agrobacterium* depends on efficient callus regeneration protocols.

Herein, we used transcriptome sequencing technology to explore the molecular mechanism of plant hormones regulating callus regeneration [8]. The transcriptome analysis during the regeneration in duckweed has not been previously studied. The growth and development of callus is mediated by many plant hormones [5]. The balance of auxin and cytokinin is the basis for *in vitro* tissue culture [9]. Explants can be incubated to callus on auxin-rich callus-inducing medium (CIM). On cytokinin-rich shoot inducing medium (SIM), the vigorous callus can be induced to de novo shoots. Hence, it is important to study the mechanism of duckweed regeneration via dynamic hormonal and transcriptional changes.

The volatile organic compounds (VOCs) could be produced to defend against herbivores, and may also play a secondary role in attracting natural enemies, which is allelopathy [10, 11]. For example, the VOCs of *Artemisia frigida Willd* play an allelopathic role on the seed germination of pasture grasses [12]. Interestingly, we found the plant regeneration could be promoted by regeneration callus. The signaling mechanisms and VOCs released from regenerating callus need to be investigated.

The main objectives of this study were: (i) identifying the molecular mechanism controlling regeneration by comprehensive transcriptomic comparison between callus and regenerating callus; (ii) identifying the VOCs that are increased during the stage of plant regeneration; (iii) examining the allelopathic effects of VOCs on the inducement of callus regeneration; (iv) conducting a transcriptome analysis on the regenerating callus promoted by VOCs.

## Results

### Promoting effect of regenerating tissue

Frond regeneration of duckweed was promoted after co-culture with regenerating callus (Co). Frond formed in 14 d with Co treatment, and duckweed regenerated at 21 days with Co treatment (Fig 1A). In the Co group, significant enhancement was found in the percentage of callus regeneration (76.8%), while the callus regeneration percentage without co-culture was 53.6% (Fig 1B, S1 Fig). Thus, the callus regeneration has been significantly increased by Co treatment.

### Transcriptome analysis identified genes, genomes and differentially expressed genes (DEGs) in regenerating callus

To compare the enriched pathways between regenerating callus (RG) and callus (CL), the Kyoto Encyclopedia of Genes and Genomes (KEGG) pathway analysis was conducted (Fig 2). The top 20 KEGG pathways with the highest representation of DEGs were analyzed. We selected the 20 pathway items that were most significant in the enrichment process. As shown in Fig 2A, the "Photosynthesis antenna proteins" was the most significantly enhanced pathway among the top 20 up-regulated KEGG pathways with the highest rich factors of RG vs. CL. This indicated that the expression of antenna protein increased after the callus developed into regenerated tissue. Antenna proteins are crucial for plant photochemical reactions and could mediate the core of plant photosynthesis. The most significantly down-regulated pathways were the "Ribosome", "Pyrimidine metabolism", "Mismatch repair", "Homologous recombination", "DNA replication" and "Base excision repair", which were among the top list of enriched pathways (Fig 2B), these were all related to the replication of DNA.

In order to understand the difference of DEGs in the regenerating callus, gene ontology (GO) enrichment analysis was conducted in RG vs. CL. As shown in Fig 2C, "cell" "cellpart"

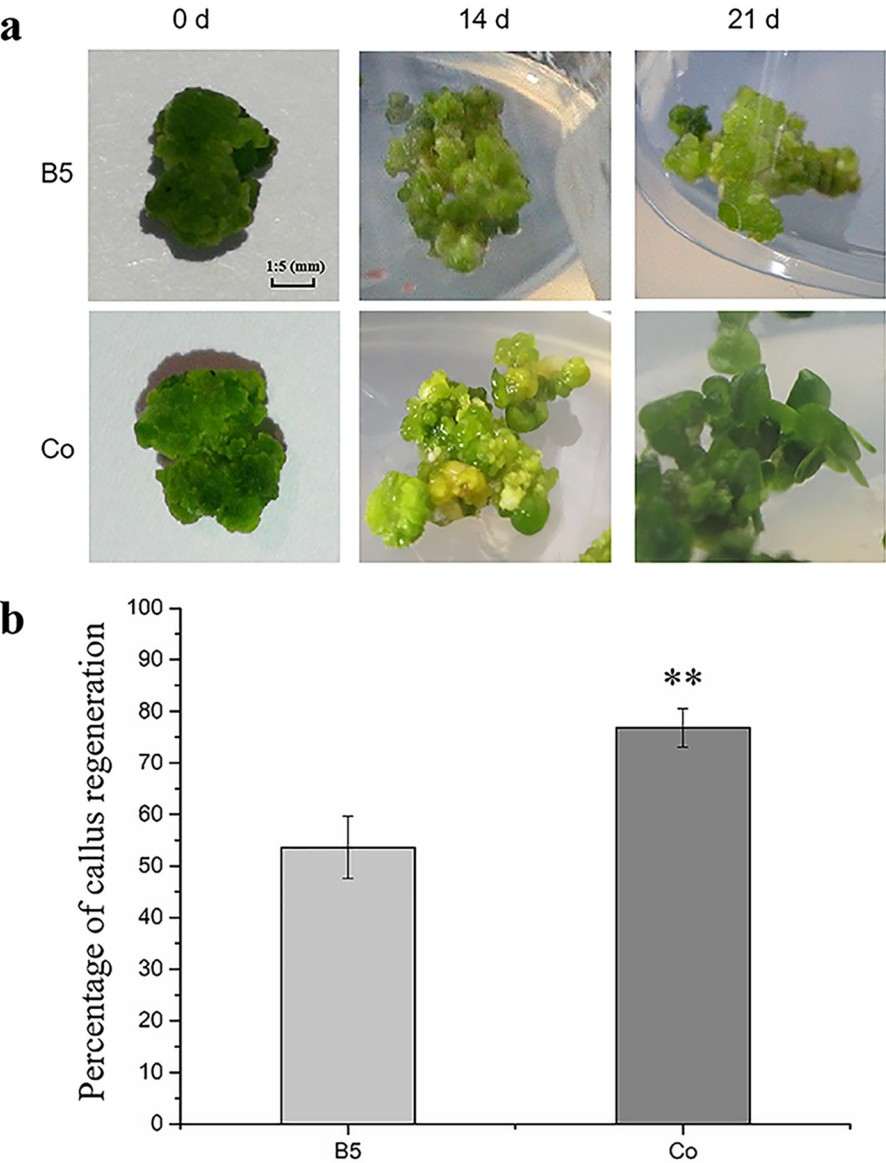

**Fig 1.** a, the co-cultured of callus and regenerating callus; Fig 1B, the callus regeneration percentage.

and "intracellular" were in biological process with the most up-regulated and down-regulated DEGs. These were followed by "macromolecular complex" and "organelle" in the category of biological process, with the most up-regulated and down-regulated DEGs. "DNA integration", "pollination", and "cell recognition" had up-regulated DEGs, but no down-regulated DEGs (Figs 2 and S2).

## Expression changes of genes related to auxin and cytokinin signaling pathway in regenerating callus

The mRNA expression was conducted by Novogene in order to study the genes that participated during callus regeneration. The course of auxin signaling pathway and related response factors are described in Fig 3A. Transport inhibitor response 1 (TIR1) and stem cell factor

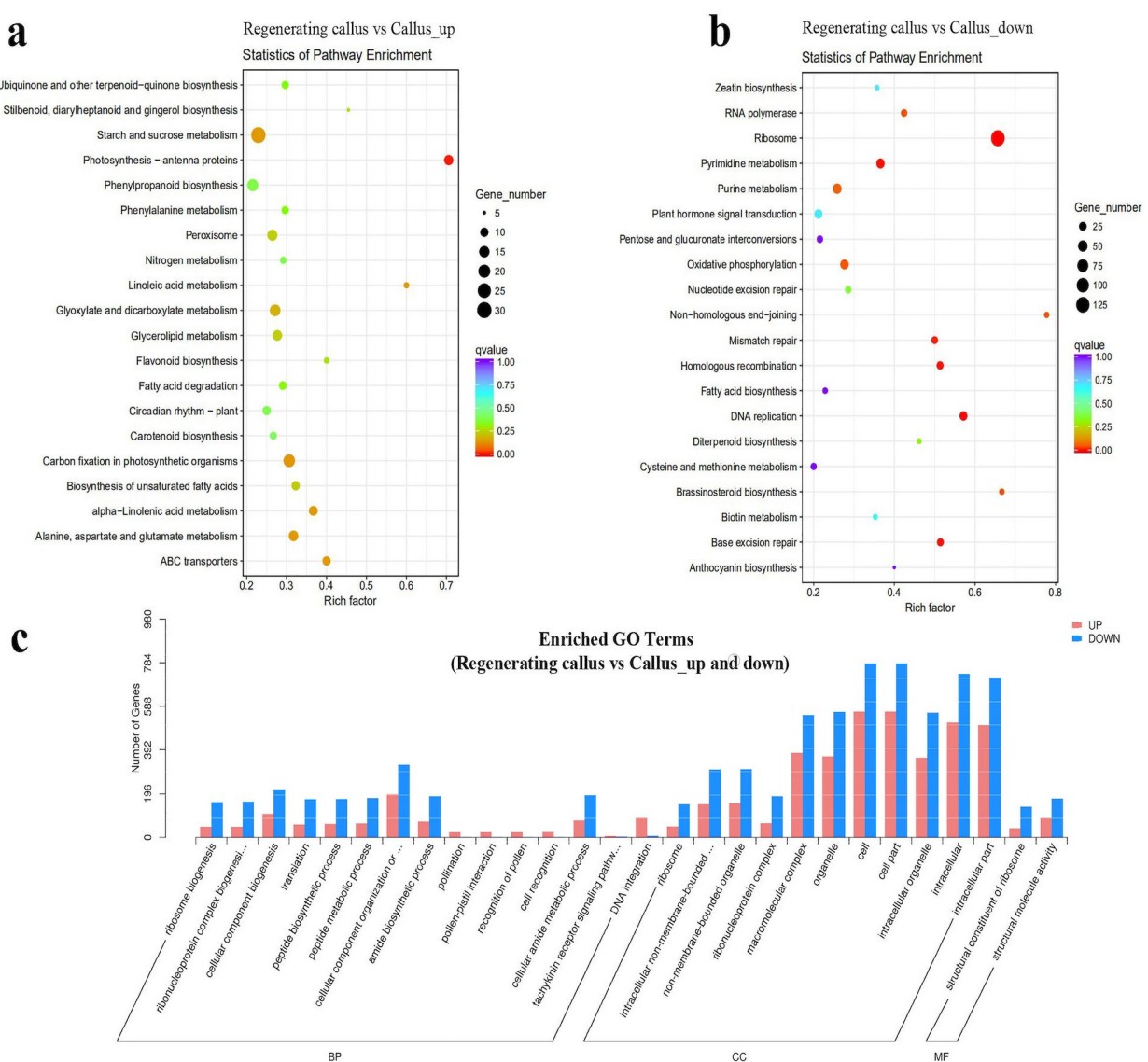

**Fig 2. Statistics of KEGG pathway enrichment and the number of enriched genes in different gene ontology (GO) categories in RG vs. CL.** Fig 2A, the most significantly enhanced pathway among the top 20 up-regulated KEGG pathways with the highest rich factors of RG vs. CL; 2b, the top list of enriched pathways; 2c, the biological process with the most up-regulated and down-regulated DEGs.

(SCF), which initiate subsequent signal transduction by binding to auxin, were down-regulated in the regenerating callus. As a transcriptional activator, auxin response factor (ARF) could regulate auxin reaction by binding to auxin-responsive protein IAA (AUX/IAA). In this study, AUX/IAA and ARF have been significantly down-regulated, by 13.03 and 3.01 $\log^2$ fold change, respectively. Auxin early response factor could be divided into three categories, which were AUX/IAA, Gretchen Hagen 3 (GH3) and small auxin-up RNA (SAUR). GH3 and SAUR were down-regulated during regeneration, as well. Ethylene-responsive factor 3 (ERF3) and Wuschel-related homeobox 11 (WOX11), which play a role in the initiation and regulation of adventitious roots (ARs), were both down-regulated. Also, lateral roots (LRs) and root hairs (RHs) relied on zinc finger protein (ZFP) and cytochrome P450 (CYP2). The expression of ZFP was decreased by 4.04 $\log^2$ fold change.

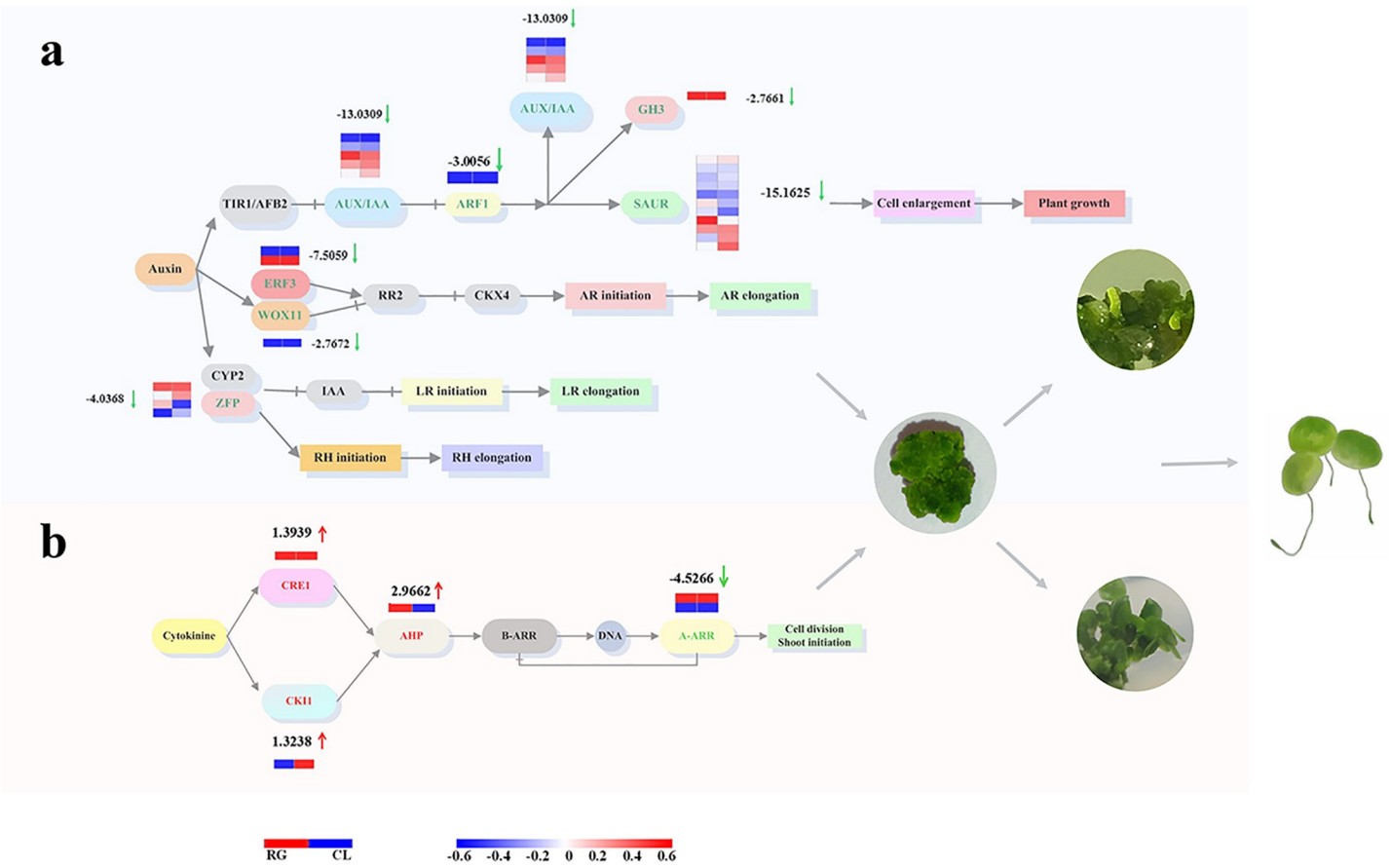

**Fig 3. a** The comparison of auxin metabolism between regenerating callus and callus is related to auxin signal transduction pathway. Transport inhibitor response 1 (TIR1), stem cell factor (SCF), auxin response factor (ARF), auxin-responsive protein IAA (AUX/IAA), Gretchen Hagen 3 (GH3), small auxin-up RNA (SAUR), Ethylene-responsive factor 3 (ERF3), Wuschel-related homeobox 11 (WOX11), adventitious roots (ARs), lateral roots (LRs), root hairs (RHs), zinc finger protein (ZFP), cytochrome P450 (CYP2). **b** Comparing regenerating callus and callus the cytokinin metabolic response and cytokinin signal transduction pathway. Cytokinin receptor 1 (CRE1), cytokinin independent 1 (CKI1), Histidine phosphate transfer protein (AHP), type-A Arabidopsis response regulators (A-ARRs), type-B Arabidopsis response regulators (B-ARR).

To identify the candidates that regulate regeneration, we studied the regulation of cytokinin signaling pathway. As shown in Fig 3B, cytokinin receptor 1 (CRE1) and cytokinin independent 1 (CKI1), as cytokinin receptors [13, 14], were up-regulated in regenerating callus. Histidine phosphate transfer protein (AHP), which interacts with CRE1 and CKI1, was up-regulated by 2.97 $\log^2$ fold change. Type-A Arabidopsis response regulators (A-ARRs) act as a negative feedback regulator, which inhibit the activity of type-B Arabidopsis response regulators (B-ARR) and form a negative feedback cycle [15, 16]. A-ARR was down-regulated by 4.53 $\log^2$ fold change, which might lead to overall up-regulation in cytokinins during callus regeneration.

## Differentially expressed genes (DEGs) between regenerating callus and callus

A total of 5,795 DEGs were found in "RG vs CL", among which 2,797 DEGs were increased, and 2,998 DEGs were decreased (Fig 4A). Through the comparative analysis of RG and CL samples, it was found that there were 20,002 DEGs shared both of them. Shared DEGs

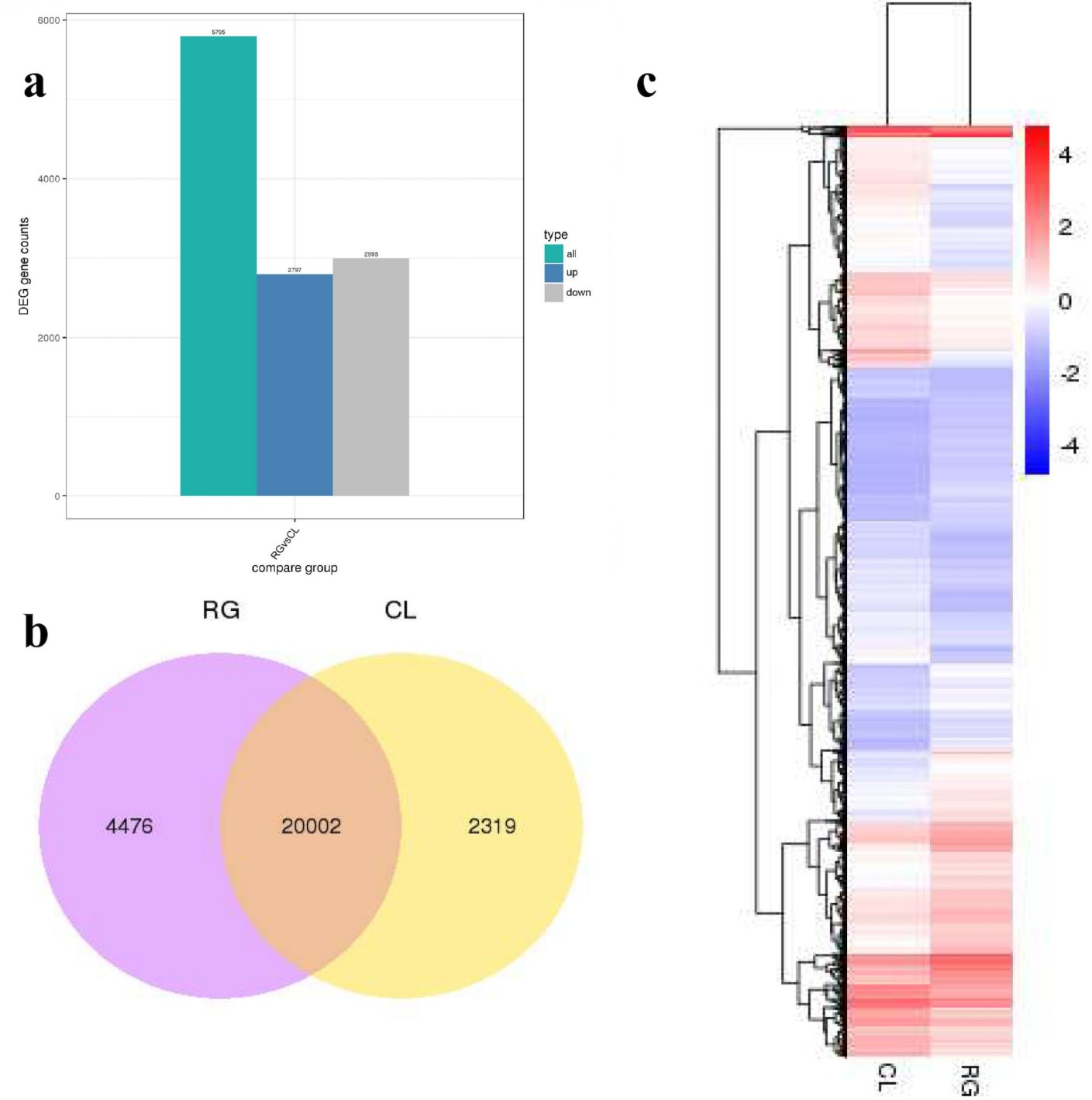

**Fig 4. Differentially expressed genes of regeneration and callus. a** Regeneration tissue compared with callus significantly up-regulated and down-regulated genes; **b** Venn diagram of DGEs in regeneration and callus; **c** Hierarchical cluster analysis showed that the total differentially expressed genes showed an upward or downward trend.

accounted for about 75% of total DEGs (Fig 4B). It suggested that the expression of major DEGs could be up-regulated or down-regulated during callus regeneration. Fig 4C, the heat map showed gene expression levels that were up-regulated and down-regulated in the case of

"regenerating callus vs callus". The redder, the higher the gene expression. The bluer, the lower the expression.

## Changes of VOCs during callus regeneration

The VOCs of regenerating callus were investigated. The qualitative and quantitative analyses of the GC/MS data were obtained from NIST/EPA/NIH Mass Spectral Library (Figs 5 and S3).

Compared to the callus, 11 types of unique VOCs in the regenerating callus were enhanced (Table 1). The peak area of 1, 3-dimethyl benzene in the regenerating callus was $0.84^*10^7$, 3.23 times than that in the callus. The emission of 1, 3-dimethyl benzene showed the highest increase in the regenerating callus. Besides, the content of 4-methyl-2-pentanol and cyclohexane also improved. Compared with the cyclohexane peak area of the callus ($0.85^*10^7$), the cyclohexane peak area of the regenerating callus was $1.28^*10^7$, 1.5 times higher than that of callus. The peak area of 4-methyl-2-pentanol was $2.1^*10^7$, 2.33 times than that of callus.

## Callus regeneration was promoted by cyclohexane

In order to explore the effect of VOCs in callus regeneration, 1, 3-dimethyl benzene, 4-methyl-2-pentanol and cyclohexane were added to the medium of callus. As shown in Fig 6, cyclohexane significantly promoted the root formation. After 16 days of cyclohexane treatment, roots formed from the callus. The newborn roots could be distinctly observed, as shown by the red arrow. However, 1, 3-dimethyl benzene and 4-methyl-2-pentanol groups showed no obvious regeneration in 16 days.

## Transcriptome analysis identified KEGGs and DEGs in callus treated by cyclohexane

Transcriptome analysis was performed to investigate the potential functions of KEGGs and DEGs in the callus treated with hydrolyzing O-glycosyl compounds by cyclohexane. As shown in Fig 7A, "RNA transport" and "glycolysis/gluconeogenesis, and galaclose metabolism" were in the biological process with the most down-regulated KEGGs. "Ribosome" was the top-enriched pathway (Rich factor >0.55). It was followed by "photosynthesis" and "oxidative phosphorylation" (Fig 7B).

In order to understand the difference of DEGs in callus treated with cyclohexane, GO enrichment analysis was conducted in callus treated by cyclohexane vs. callus. As shown in Fig 7C, "DNA integration", "ribonucleoprotein complex" and "structural molecule activity" were in the biological process with the most up-regulated DEGs. These were followed by "ribosome biogenesis", "ribonucleoprotein complex" and "ribosome" in the category of biological process with the most up-regulated DEGs. Meanwhile, "ribonucleoprotein complex" and "structural molecule activity" were in the biological process with the most down-regulated DEGs (Figs 7C and S4).

## Comparison of the expression of genes related to hormones in callus treated with cyclohexane and in the regenerating callus

In order to identify the molecular factors underlying the participation of hormones in callus regeneration, we first checked gene expression related to auxin signaling pathway (Table 2). AUX/IAA and GH3 were down regulated in both callus treated with cyclohexane and in the regenerating callus. A majority of SAUR were down-regulated during regeneration and treatment with cyclohexane (Fig 8A). ERF3, cysteine-rich receptor and zinc finger were down-regulated as well.

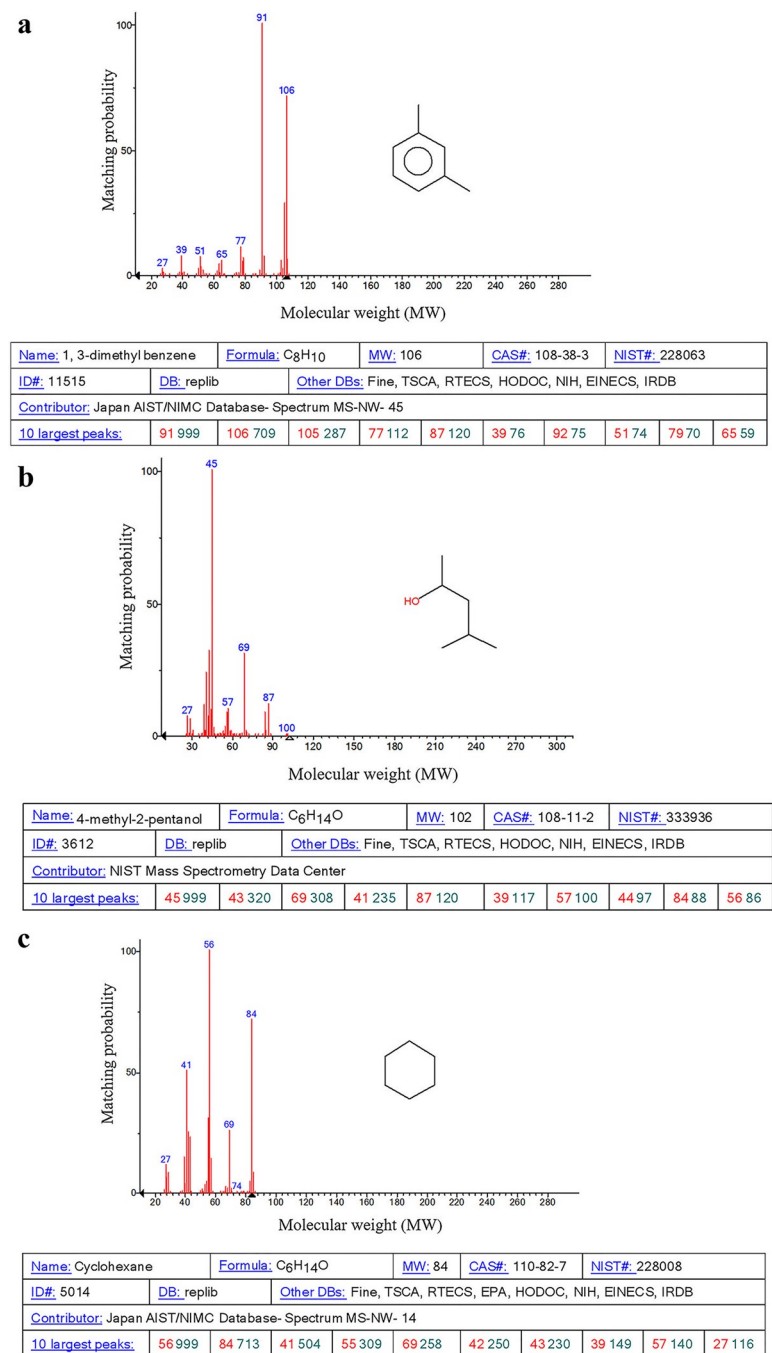

**Fig 5. Three types of VOCs were significantly up-regulated in the callus regeneration stage.** The numbers in blue represented the mass-to-charge ratio (m/z) of a substance in the histogram.

Second, we studied the expression of genes related to CTK signaling (Fig 8B). The gene regulation in regeneration and treatment with cyclohexane is different. The CRE1 was up-regulated in the regenerating callus, and down-regulated in callus treated with cyclohexane (Table 3).

Third, the expression of genes related to brassionosteroid signal were investigated. In the brassionosteroid signaling pathway, the expression of brassinazole-resistant 1/2 (BZR1/2) was

**Table 1. The main components of VOCs from regenerating callus and callus.**

| Designation | Chemical formula | RG Peak area ($*10^7$) | CL Peak area ($*10^7$) | Acquisition time (min) |
|---|---|---|---|---|
| Cyclohexane | $C_6H_{12}$ | 1.28 | 0.85 | 3.06 |
| 9,12, 15-octadecarboxylic acid methyl ester | $C_{28}H_{40}O_4$ | 0.44 | 0.4 | 3.32 |
| 10,13-octadecadiynoic acid methyl ester | $C_{19}H_{30}O_2$ | 3.49 | 3.3 | 3.38 |
| 4-methyl-2-pentanol | $C_6H_{14}O$ | 2.1 | 0.9 | 3.81 |
| 1, 3-dimethyl benzene | $C_8H_{10}$ | 0.84 | 0.26 | 5.83 |
| 1,1'-oxybis-decane | $C_{20}H_{42}O$ | 0.95 | 0.48 | 15.82 |
| Diisobutyl phthalate | $C_{26}H_{44}O_5$ | 1.88 | 1.75 | 17.17 |
| Nonadecane | $C_{19}H_{40}$ | 0.8 | 0.64 | 19.15 |
| 3-(2,6,6-trimethyl-1-cyclohexen-1-yl)-2-propenal | $C_{12}H_{18}O$ | 1.28 | 0.9 | 24.13 |
| 9,10-dihydro-11,12-diacetyl-9,10-ethanoanthracene | $C_{20}H_{18}O_2$ | 2.75 | 1.8 | 31.81 |
| Butyl 8-methylnonyl ester 1,2-benzenedicarboxylic acid | $C_{22}H_{34}O_4$ | 1.21 | 0.79 | 34.2 |

down-regulated in callus treated with cyclohexane and the regenerating callus (Table 4). In the brassionosteroid signaling pathway, the expression of BZR1/2 was down-regulated in callus treated with cyclohexane and the regenerating callus.

Moreover, the expression of genes related to ethylene signaling were investigated (Fig 8C). The expression of ETR and EBF1/2 were up-regulated in callus treated with cyclohexane and the regenerating callus. Transcription factor MYC2, which plays a role in jasmonic acid signaling pathway, was up-regulated after cyclohexane treatment and in regenerating callus (Fig 8D). There was no significant difference in gibberellin signaling pathway during cyclohexane treatment (Fig 8E). The expression of genes related to plant hormones was showen as S5 Fig.

## Discussion

In accordance with previous studies, we established an effective way for *in vitro* callus regeneration in duckweed. Interestingly, we found that a regenerating callus promoted another callus to regenerate. Transcriptome sequencing (especially plant hormones) and volatile substances were studied to reveal the molecule framework of plant regeneration in duckweed.

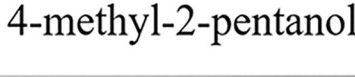
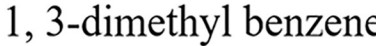
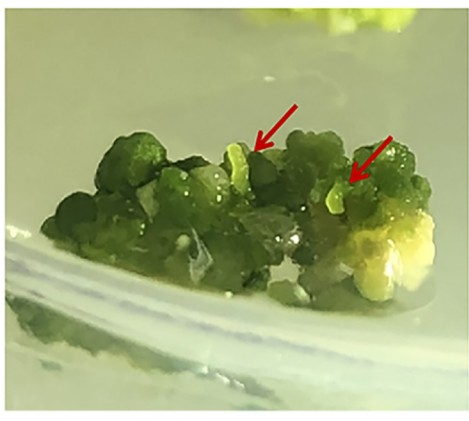
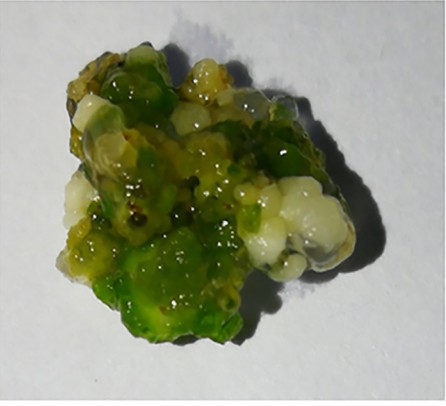
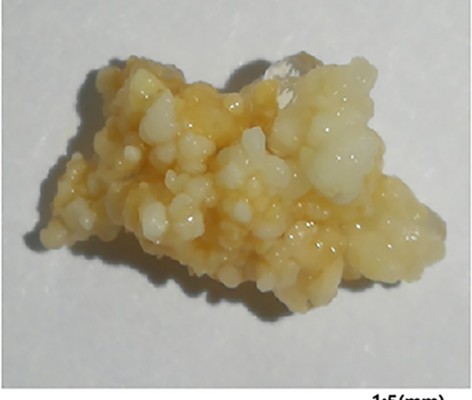

**Fig 6. Effects of treatment of callus with three VOCs (cyclohexane, 4-methyl-2-pentanol and 1, 3-dimethyl benzene) for 16 days.**

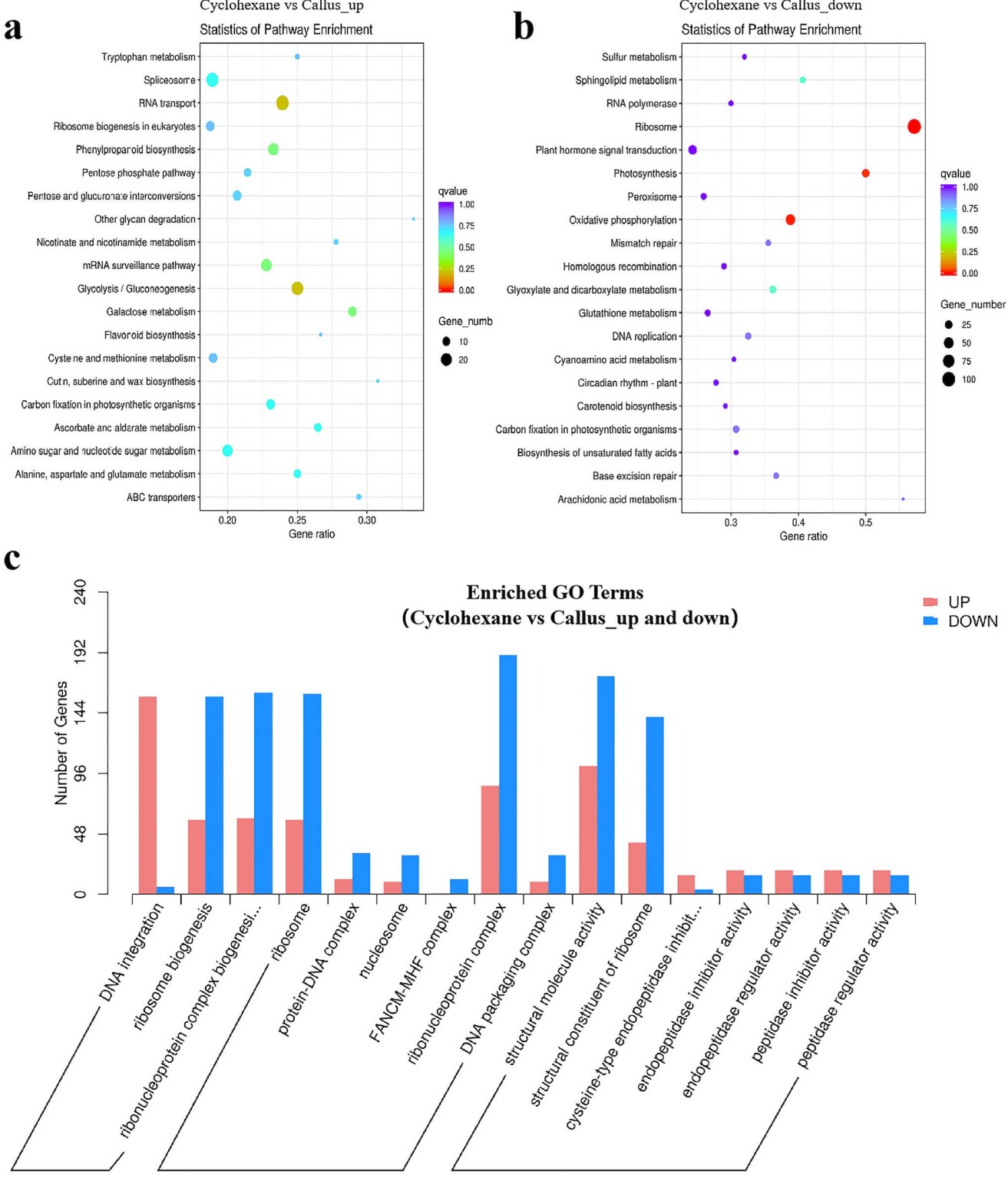

**Fig 7.** The top 20 up-regulated (Fig 7A) and down-regulated (Fig 7A) DEGs in KEGG pathways of "Cyclohexane vs. CL"; Fig 7C, the enriched GO terms.

**Table 2. Gene expression during plant regeneration in Auxin.**

| Description | Gene-id | Regenerating callus vs Callus_Read_count | Cyclohexane vs Callus_Read_count | Callus_Read_count | Regenerating callus vs Callus_log$^2$ Fold Change | Cyclohexane vs Callus_log$^2$ Fold Change | pval | padj |
|---|---|---|---|---|---|---|---|---|
| auxin-responsive protein IAA | Cluster-6172.2761 | 25.79 | / | 291.87 | -3.5 | / | 1.53E-20 | 7.20E-19 |
| auxin-responsive protein IAA | Cluster-6172.9506 | 1350.90 | / | 7436.54 | -2.46 | / | 8.50E-33 | 1.36E-30 |
| auxin-responsive protein IAA | Cluster-6172.9484 | 3097.05 | / | 8093.56 | -1.39 | / | 9.12E-09 | 8.99E-08 |
| auxin-responsive protein IAA | Cluster-6172.6741 | 115.19 | / | 752.97 | -2.72 | / | 4.06E-30 | 5.09E-28 |
| auxin-responsive protein IAA | Cluster-6172.4574 | 329.74 | / | 2581.01 | -2.97 | / | 5.94E-28 | 5.78E-26 |
| auxin-responsive protein IAA | Cluster-7966.13997 | / | 126.90 | 427.80 | / | -1.76 | 1.96E-13 | 1.49E-12 |
| auxin-responsive protein IAA | Cluster-7966.10326 | / | 2912.32 | 6803.46 | / | -1.22 | 1.54E-20 | 1.99E-19 |
| auxin-responsive protein IAA | Cluster-7966.9984 | / | 757.43 | 2282.82 | / | -1.59 | 5.19E-39 | 2.10E-37 |
| auxin-responsive protein IAA | Cluster-7966.7990 | / | 882.15 | 7136.37 | / | -3.02 | 1.35E-109 | 9.48E-107 |
| auxin-responsive protein IAA | Cluster-7966.3823 | / | 24.87 | 135.71 | / | -2.45 | 3.10E-16 | 2.89E-15 |
| auxin-responsive protein IAA | Cluster-7966.9412 | / | 68.99 | 688.29 | / | -3.32 | 1.37E-77 | 3.22E-75 |
| auxin-responsive protein IAA | Cluster-7966.8499 | / | 2134.92 | 8945.72 | / | -2.07 | 4.70E-93 | 1.83E-90 |
| auxin response factor | Cluster-6172.11643 | 642.33 | / | 5159.89 | -3.01 | / | 1.02E-27 | 9.71E-26 |
| auxin response factor | Cluster-7966.6357 | / | 821.41 | 2005.68 | / | -1.29 | 1.79E-22 | 2.67E-21 |
| auxin response factor | Cluster-7966.4925 | / | 2164.10 | 4677.23 | / | -1.11 | 8.03E-30 | 1.93E-28 |
| auxin-responsive GH3 gene family | Cluster-6172.10088 | 766.11 | / | 5210.20 | -2.77 | / | 7.72E-22 | 4.27E-20 |
| auxin-responsive GH3 gene family | Cluster-7966.4925 | / | 2164.10 | 4677.23 | / | -1.11 | 8.03E-30 | 1.93E-28 |
| SAUR family protein | Cluster-6172.1833 | 1482.63 | / | 191.08 | 2.96 | / | 1.56E-18 | 5.86E-17 |
| SAUR family protein | Cluster-6172.15713 | 151.85 | / | 76.04 | 1 | / | 0.0052791 | 0.017525 |

(*Continued*)

**Table 2.** (Continued)

| Description | Gene-id | Regenerating callus vs Callus_Read_count | Cyclohexane vs Callus_Read_count | Callus_Read_count | Regenerating callus vs Callus_log$^2$ Fold Change | Cyclohexane vs Callus_log$^2$ Fold Change | pval | padj |
|---|---|---|---|---|---|---|---|---|
| SAUR family protein | Cluster-2913.0 | 88.15 | / | 25.02 | 1.82 | / | 1.24E-05 | 7.06E-05 |
| SAUR family protein | Cluster-3967.0 | 0.34 | / | 9.71 | -4.74 | / | 0.0020559 | 0.007526 |
| SAUR family protein | Cluster-6172.19466 | 95.93 | / | 263.01 | -1.45 | / | 0.00046131 | 0.0019407 |
| SAUR family protein | Cluster-6172.1791 | 33.88 | / | 139.87 | -2.06 | / | 1.44E-09 | 1.61E-08 |
| SAUR family protein | Cluster-6172.18366 | 200.57 | / | 541.46 | -1.43 | / | 3.18E-08 | 2.84E-07 |
| SAUR family protein | Cluster-6172.17182 | 61.02 | / | 123.87 | -1.03 | / | 0.0078308 | 0.024821 |
| SAUR family protein | Cluster-6172.17013 | 11.05 | / | 235.41 | -4.44 | / | 6.48E-30 | 7.95E-28 |
| SAUR family protein | Cluster-5374.0 | 11.48 | / | 33.53 | -1.52 | / | 0.016176 | 0.046631 |
| SAUR family protein | Cluster-6172.13654 | 51.66 | / | 993.51 | -4.26 | / | 8.32E-34 | 1.46E-31 |
| SAUR family protein | Cluster-1875.0 | / | 191.22 | 56.28 | / | 1.77 | 1.48E-12 | 1.04E-11 |
| SAUR family protein | Cluster-7966.1555 | / | 9.76 | 60.96 | / | -2.64 | 4.36E-07 | 1.89E-06 |
| SAUR family protein | Cluster-3489.0 | / | 26.37 | 109.08 | / | -2.05 | 1.81E-11 | 1.17E-10 |
| SAUR family protein | Cluster-7372.0 | / | 50.62 | 235.25 | / | -2.21 | 5.44E-15 | 4.64E-14 |
| SAUR family protein | Cluster-7966.7594 | / | 163.41 | 768.46 | / | -2.24 | 2.29E-31 | 6.09E-30 |
| SAUR family protein | Cluster-7966.11015 | / | 217.56 | 523.54 | / | -1.27 | 5.94E-18 | 6.34E-17 |
| SAUR family protein | Cluster-7966.4605 | / | 222.98 | 490.60 | / | -1.14 | 7.22E-07 | 3.04E-06 |
| SAUR family protein | Cluster-7966.15997 | / | 23.92 | 206.53 | / | -3.12 | 4.05E-27 | 8.18E-26 |
| SAUR family protein | Cluster-7966.11607 | / | 88.77 | 876.28 | / | -3.31 | 1.04E-55 | 9.68E-54 |
| Ethylene-responsive transcription factor 3 | Cluster-6172.9509 | 96.48 | / | 1004.22 | -3.38 | / | 1.88E-29 | 2.15E-27 |
| Ethylene-responsive transcription factor 3 | Cluster-6172.14530 | 97.26 | / | 1695.50 | -4.12 | / | 1.27E-35 | 2.68E-33 |
| cysteine-rich receptor | Cluster-6172.505 | 11.82 | / | 80.91 | -2.77 | / | 1.41E-06 | 9.59E-06 |
| Zinc finger | Cluster-6172.2152 | 66.61 | / | 133.41 | -1 | / | 0.011746 | 0.035246 |
| Zinc finger | Cluster-6172.19271 | 48.99 | / | 12.35 | 2 | / | 0.00093637 | 0.0036929 |

(*Continued*)

**Table 2.** (Continued)

| Description | Gene-id | Regenerating callus vs Callus_Read_count | Cyclohexane vs Callus_Read_count | Callus_Read_count | Regenerating callus vs Callus_log² Fold Change | Cyclohexane vs Callus_log² Fold Change | pval | padj |
|---|---|---|---|---|---|---|---|---|
| Zinc finger | Cluster-2307.0 | 24.64 | / | 90.13 | -1.86 | / | 0.00013142 | 0.0006126 |
| Zinc finger | Cluster-2857.0 | 1.30 | / | 11.91 | -3.17 | / | 0.0031694 | 0.011129 |

Plant hormones play a crucial role during callus regeneration [1]. The aim of the present study was to have a deeper understanding of the regulatory mechanism of callus regeneration. Callus was induced by auxin, similar to lateral root primordium [17–19]. In Arabidopsis, the callus tissue formed root stem cell niche, by regulation the expression of root stem cell regulators, including WOX [20–23]. The gene of ARF, AUX/IAA, GH3, ARF1, SAUG and other response factors were significantly down expressed during callus redifferentiation (Fig 3). In the auxin signaling pathway, the interaction between ARF and AUX /IAA could regulate the gene expression of auxin early response. Moreover, ERF3, WOX11 and ZFP were found to be related to the ARs, LRs and RHs of initiation in *Spirodela* [8], which might be associated with the regeneration in duckweed.

Cytokinins and auxin have synergistic or antagonistic interactions with each other [24]. As a phytohormone, cytokinin controls key aspects of environmental responses, such as biotic and abiotic stress responses, as well as regulates various developmental processes including cell proliferation, leaf formation, and root formation and growth [25, 26]. Cytokinins promote plant regeneration by regulating the generation of somatic embryogenesis in *Fumariaceae* and rice [27, 28]. In our study, cytokinin receptor CRE1, CKI1 and transfer protein of histidine phosphate AHP were enhanced. The expression of cytokinins was up-regulated, thereby promoting the differentiation of shoots. The transcriptome analysis showed similar result with Arabidopsis, providing evidence that the regulation of auxin and cytokinins leads to regeneration. Besides, plant regeneration is regulated by other hormones [29]. We found that gibberellin and jasmonic acid increased significantly, while genes related to brassinolide was downregulated during callus regeneration.

Plants release VOCs into the environment to affect their own or other biological life processes in the process of growth and development. This phenomenon is called allelopathy [30]. Plants growing under biological stress or abiotic stress might release different VOCs to improve their resistance to external interference [31–33]. VOCs have been shown to mediate cell to cell communication, thereby leading to stress responses in plants [34]. Here, we found that cyclohexane could significantly promote the regeneration of callus in 16 days, discover the role of VOC during callus regeneration.

The regulation of gene expression related to hormones in callus treated with cyclohexane, which promoted regeneration, suggested the role of auxin during regeneration. AUX/IAA and GH3 have been down regulated in callus treated with cyclohexane, which was similar to that in the regenerating callus (Fig 8). Adventitious root initiation and elongation were promoted by AUX/IAA [8]. Interestingly, the root formation was significantly enhanced by cyclohexane treatment in our results.

Taken together, we propose a hypothesis how callus regenerates in duckweed. Allelopathy affects duckweed callus regeneration. Cyclohexane released from regenerating callus triggers root formation via hormonal transcriptional regulation, especially with a down-regulation of auxin relation genes. Cyclohexane treatment enhanced duckweed regeneration by initiating

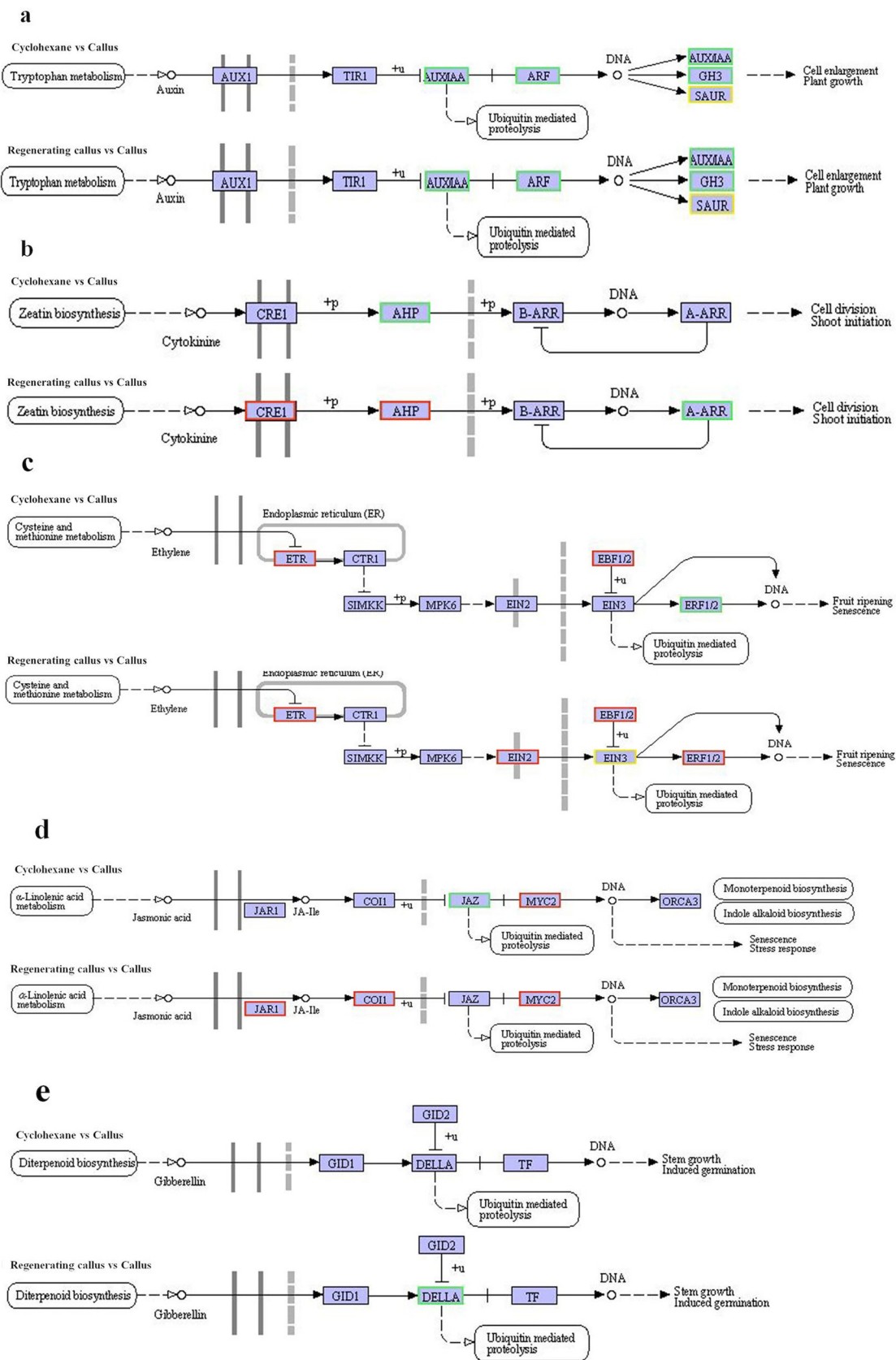

**Fig 8. The expression of genes related to hormones in callus treated with cyclohexane and in the regenerating callus.** Red indicated high expression, and blue indicated low expression.

root. The results indicated that VOCs might play a crucial role in the process of plant regeneration.

## Materials and methods

### Plant material, *in vitro* establishment and cyclohexane treatment

Duckweed (*Lemna turionifera*) were collected from Fengchan River, Xiqing District, Tianjin (117.1˚E longitude, 39.1˚N latitude), China. Duckweed was aseptically cultured in liquid medium as previously described [35, 36]. Fully expanded fronds were selected as explant for callus induction. The rhizoid was removed, and the frond was scratched for callus induction. The induction medium was B5 solid medium, which was designed by Gamborg for soybean tissue culture in 1968 [37]. The induction medium contained plant hormones 15 mg L$^{-1}$ dicamba, 3.5 mg L$^{-1}$ 2,4dichlorophenoxy acetic acid (2, 4-D), 2 mg L$^{-1}$ 6-benzylaminopurine (6-BA) and 1.5% sucrose. The pH of the medium was adjusted to 6.2–6.4 prior to addition of 0.6% agargel, followed by sterilization at 121˚C for 20 minutes. The tissue was maintained in an incubator at 23 ± 2˚C, with a light cycle of 16 hours light and 8 hours darkness. After 4–5 weeks of induction, the duckweed explants developed into callus through dedifferentiation.

After 2–3 weeks of induction, calli were formed (diameter about 4 mm). The calli were transferred to the subculture medium containing B5 medium, 10 mg L$^{-1}$ 4-chlorophenoxyace-tic acid (CPA) and 2 mg L$^{-1}$ 6-c-c-(dimethylallylamino)-purine (2Ip). In order to maintain the morphology and activity of the callus, the subculture medium was replaced every two weeks.

**Table 3. Gene expression during plant regeneration in Cytokinin.**

| Description | Gene-id | Regenerating callus vs Callus_Read_count | Cyclohexane vs Callus_Read_count | Callus_Read_count | Regenerating callus vs Callus_log2Fold Change | Cyclohexane vs Callus_log2Fold Change | pval | padj |
|---|---|---|---|---|---|---|---|---|
| cytokinin receptor (arabidopsis histidine kinase 2/3/4) | Cluster-6172.61 | 7743.07 | / | 2946.79 | 1.39 | / | 4.11E-13 | 7.79E-12 |
| histidine-containing phosphotransfer peotein | Cluster-6172.20 | 165.19 | / | 21.04 | 2.97 | / | 3.08E-13 | 5.97E-12 |
| histidine-containing phosphotransfer protein | Cluster-7966.45 | / | 264.88 | 801.41 | / | -1.60 | 3.14E-23 | 4.90E-22 |
| histidine-containing phosphotransfer protein | Cluster-2808.0 | / | 3.44 | 19.25 | / | -2.46 | 0.0049514 | 0.012128 |
| two-component response regulator ARR-A family | Cluster-6172.13 | 118.81 | / | 737.90 | -2.63 | / | 4.93E-15 | 1.22E-13 |
| two-component response regulator ARR-A family | Cluster-4229.0 | 14.43 | / | 54.48 | -1.90 | / | 0.0091425 | 0.028407 |
| Histidine kinase CKI1 | Cluster-6172.41 | 765.06 | / | 305.76 | 1.32 | / | 5.13E-10 | 6.17E-09 |

**Table 4. Gene expression during plant regeneration in Brassinosteroid.**

| Description | Gene-id | Regenerating callus vs Callus_Read_count | Cyclohexane vs Callus_Read_count | Callus_Read_count | Regenerating callus vs Callus_log2 Fold Change | Cyclohexane vs Callus_log2 Fold Change | pval | padj |
|---|---|---|---|---|---|---|---|---|
| BRI1 kinase inhibitor 1 | Cluster-6172.8113 | 291.51 | / | 769.98 | -1.40 | / | 3.62E-07 | 2.73E-06 |
| brassinosteroid resistant 1/2 | Cluster-6172.9208 | 243.31 | / | 545.79 | -1.17 | / | 3.52E-07 | 2.66E-06 |
| brassinosteroid resistant 1/2 | Cluster-6401.0 | / | 43.04 | 146.14 | / | -1.77 | 9.00E-11 | 5.52E-10 |
| brassinosteroid resistant 1/2 | Cluster-6172.20298 | 33.13 | / | 156.76 | -2.24 | / | 5.50E-07 | 4.01E-06 |
| cyclin D3 | Cluster-6172.6746 | 932.34 | / | 2811.63 | -1.59 | / | 3.80E-21 | 1.91E-19 |

Callus was transferred to the regeneration medium for duckweed regeneration. The regeneration medium contained B5 medium, 1 mM serine, and 1.5% sucrose. After 2 or 3 weeks, the callus re-differentiated fronds with rootlets.

Cyclohexane treatment: After 3 days culture in B5 subculture medium, the calli were cultured in B5 medium with 20 ml cyclohexane in a large airtight beaker. The sealing device was regularly opened every day to change the air in the beaker. Cyclohexane was replaced every two days.

## The co-culture of regenerating callus and callus

The callus was cultured on subculture medium for more than two weeks for subsequent experiments. Callus and regenerating callus in the same growth condition were placed in B5 medium (containing 1.5% sucrose), respectively. For fumigate, the regenerating callus and callus were placed together in a closed environment for co-culture (Fig 9A).

## VOC collection and analysis

As shown in Fig 9B, the VOCs released from callus and regenerating duckweed were collected using the dynamic headspace air-circulation method described by Zuo et al. (2018) [38]. There were three conical flasks of callus or regenerating callus for each group, each bottle of tissue weighs 1.3–1.5g. The chemical composition analysis of VOCs was performed by thermal-desorption system/gas chromatography/mass spectrum (TDS/GC/MS). The GC/MS data was studied in NIST/EPA/NIH Mass Spectral Library (NIST 08) (National Institute of Standards and Technology, MD, USA).

## RNA isolation, quantification, and sequencing

RNA samples from duckweed callus were extrated by Tiangen kit (Tiangen RNAsimple total rna kit). mRNA was purified from total RNA using poly-T oligo-attached magnetic beads. RNA concentration was measured using Qubit® RNA assay kit in Qubit® 2.0 Flurometer (Life Technologies, CA, USA). RNA integrity was assessed using the RNA Nano 6000 assay kit and the Agilent Bioanalyzer 2100 system (Agilent Technologies, CA, USA). After that, clustering and sequencing index-coded samples clustering has been performed. After passing the library inspection, according to the requirements of effective concentration and target offline data volume, the Illumina sequencing was performed after pooling.

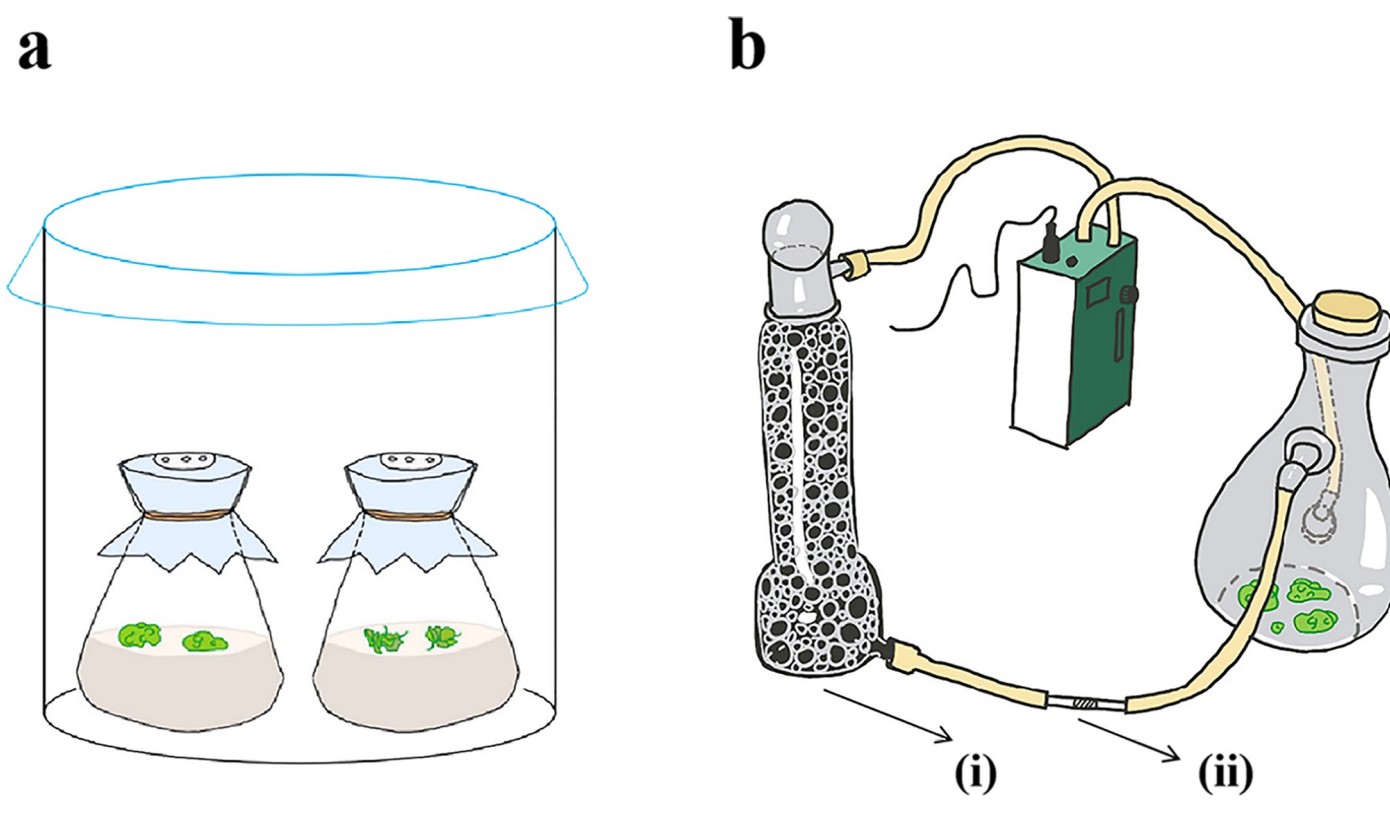

**Fig 9.**

## Sequencing, data filtering, transcript assembly and analyse

Image data from sequencing fragments measured by high-throughput sequencers were transformed into sequence data (reads) by CASAVA base recognition. The raw data obtained from sequencing included a small number of reads with sequencing adaptors or low sequencing quality. The filtering contents were as per our previous study: Removed adapters; removed reads whose proportion of N was >10%; removed low-quality reads [6]. The clean reads were assembled by the trinity de novo assembly program with min_kmer_cov set to 2 by default, otherwise it was set to default [39]. Overall, a reference sequence, with an average length of 1928 bp and a total length of 282527137 bp, was obtained for subsequent analysis.

Gene Ontology (GO) enrichment analysis of differentially expressed genes was implemented by the clusterProfiler R package, in which gene length bias wascorrected. GO terms with corrected Pvalue less than 0.05 were considered significantly enriched by differential expressed genes. KEGG is a database resource for understanding high-level functions and utilities of the biological system, from molecular-level information, especially large-scale molecular datasets generated by genome sequencing and other high-through put experimental technologies. We used clusterProfiler R package to test the statistical enrichment of differential expression genes in KEGG pathways.

Differential expression analysis of two conditions/groups (two biological replicates per condition) was performed using the DESeq2 R package (1.20.0). DESeq2 provide statistical routines for determining differential expression in digital gene expression data using a model based on the negative binomial distribution. The resulting P-values were adjusted using the Benjamini and Hochberg's approach for controlling the false discovery rate. Genes with an adjusted P-value <0.05 found by DESeq2 were assigned as differentially expressed.

## Data analysis

The experiment were repeated at least three independent times. Analysis of variance (ANOVA) method and SPSS software (IBM SPSS Statistics, Version 20) were applied to compare the statistical significance. Significant difference was indicated by asterisks ($^*p < 0.05$, $^{**}p < 0.01$). Standard deviations were shown by error bars. The graphs were made using Origin 9.0 (Origin Lab, USA).

## Supporting information

**S1 Fig. The record for callus regeneration percentage.** CK, control treatment; Co, the callus co-cultured with regenerating callus.
(TIFF)

**S2 Fig. KEGG enrichment scatter plot.** The 20 pathways with the most signifcant enrichment of 'RG vs CL'.
(TIFF)

**S3 Fig. The GC-MS chromatogram of VOCs.** a Data analyze of 1, 3-dimethyl benzene. b Data analyze of 4-methyl-2-pentanol. c Data analyze of cyclohexane.
(TIFF)

**S4 Fig. Diference genes GO and KEGG enrichment histogram.** a The 20 pathways with the most signifcant enrichment of 'CRvs CL'. b Histogram of GO enrichment of diferential genes in 'CR vs CL' with the most down-regulated DEGs. c Histogram of GO enrichment of diferential genes in 'CR vs CL' with the most up-regulated DEGs.
(TIFF)

**S5 Fig. The expression of genes related to plant hormones signal transduction.** a The expression of genes related to hormones in regenerating callus vs callus. b The expression of genes related to hormones in callus treated with cyclohexane vs callus.
(TIFF)

## Author Contributions

**Data curation:** Jinge Sun, Yaya Wang.

**Formal analysis:** Lin Yang.

**Investigation:** Jinge Sun, Yaya Wang.

**Methodology:** Congyu Yan, Qiuting Ren.

**Project administration:** Jinsheng Sun.

**Resources:** Lin Yang.

**Software:** Congyu Yan, Shen Wang, Xu Ma.

**Supervision:** Lin Yang, Jinsheng Sun.

**Validation:** Xu Ma, Ling Zhao.

**Visualization:** Junyi Wu, Shen Wang.

**Writing – original draft:** Lin Yang, Jinge Sun.

**Writing – review & editing:** Lin Yang, Jinge Sun, Yaya Wang.

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
