## [Decision Letter · Decision Letter 0]

21 Jun 2021

PONE-D-21-17629

The framework of plant regeneration in duckweed (Lemna turonifera) comprises genetic transcript regulation and cyclohexane release

PLOS ONE

Dear Dr. Sun,

Thank you for submitting your manuscript to PLOS ONE. After careful consideration, we feel that it has merit but does not fully meet PLOS ONE’s publication criteria as it currently stands. Therefore, we invite you to submit a revised version of the manuscript that addresses the points raised during the review process.

We look forward to receiving your revised manuscript.

Kind regards,

Jen-Tsung Chen, Ph.D.

Academic Editor

PLOS ONE

Journal Requirements:

2. In your Methods section, please specify the name of the lake (and geographic coordinates if relevant) where you collected the plant material.

"Present research has been supported by National Natural Science Foundation of China

(No. 32071620 ), Tianjin Science and technology project (19ZYPTSN00030)."

"Funded studies

Initials of the authors who received each award: L.Yang. , JG. Sun

Grant numbers awarded to each author: 86185 dollars, 785 dollars

The full name of each funder: Present research has been supported by National Natural Science Foundation of China (No. 32071620), Tianjin Science and technology project (19ZYPTSN00030)

URL of each funder website: N/A

Did the sponsors or funders play any role in the study design, data collection and analysis, decision to publish, or preparation of the manuscript?

Yes."

6. Please upload a new copy of Figure 2 and 7 as the detail is not clear. Please follow the link for more information: https://blogs.plos.org/plos/2019/06/looking-good-tips-for-creating-your-plos-figures-graphics/" https://blogs.plos.org/plos/2019/06/looking-good-tips-for-creating-your-plos-figures-graphics/

7. We note that Figures 1,6 and 9 in your submission contain copyrighted images. All PLOS content is published under the Creative Commons Attribution License (CC BY 4.0), which means that the manuscript, images, and Supporting Information files will be freely available online, and any third party is permitted to access, download, copy, distribute, and use these materials in any way, even commercially, with proper attribution. For more information, see our copyright guidelines: http://journals.plos.org/plosone/s/licenses-and-copyright.

a. You may seek permission from the original copyright holder of Figure(s) [#] to publish the content specifically under the CC BY 4.0 license. 

8. We note you have included a table to which you do not refer in the text of your manuscript. Please ensure that you refer to Table 3 in your text; if accepted, production will need this reference to link the reader to the Table.

Reviewers' comments:

Reviewer's Responses to Questions

**Comments to the Author**

1. Is the manuscript technically sound, and do the data support the conclusions?

Reviewer #1: Partly

Reviewer #2: No

Reviewer #3: Yes

2. Has the statistical analysis been performed appropriately and rigorously? 

Reviewer #1: Yes

Reviewer #2: N/A

Reviewer #3: Yes

3. Have the authors made all data underlying the findings in their manuscript fully available?

Reviewer #1: No

Reviewer #2: No

Reviewer #3: No

4. Is the manuscript presented in an intelligible fashion and written in standard English?

Reviewer #1: No

Reviewer #2: No

Reviewer #3: No

5. Review Comments to the Author

Reviewer #1: This study provides the new insights related to duckweed regeneration. Unfortunately, the results have not been well presented and have not been interpreted correctly. Also, manuscript needs to be edited by a native English editor.

Other my comments:

- Please revise abstract to highlight hypothesis/objective of work

- I recommend revising Abstract. Content is not connected.

- Lines 77-78: please revise it (e.g. double "with")

- Line 83: Transcriptome analysis identifies Genes and Genomes (KEGG): KEGG is database for pathway analysis, not abbreviation of " Transcriptome analysis identifies Genes and Genomes".

- Lines 89-91: this sentence should be rewritten. It is unclear.

- Line 94: "Pathways were" instead of "pathway was".

- Line 190: "AUX/IAA and GH3 has " it should be "have"

- Authors used a lot of "have/has been up/down-regulated" I do not recommend expressing results in this way.

- Line 202: double "in"

- In discussion, the results repeated. Please interpret the key results.

- Line 260: "has" should be changed to "have".

- Line 274: it needs to edit. Why "were"

- Lines 292-295: Unclear, please re-write them.

- Materials and methods are not perfect and more details have to add.

- How to analyze pathways, and selecting DEGs.

- Line 312: How were RNA samples extracted? Which method or kit?

Reviewer #2: In this study, the authors duckweed regeneration molecular mechanism by regenerating callus. The authors identified that mainly auxin, cytokinin and VOCs related genes and compounds play a key role in callus regeneration. Overall, there is no major issue. However, the current version is not acceptable in its present form and demands improvement. Some of the suggestions are as follows:

The unit is not consistent throughout the text. Please follow the standard unit style, i.e., mg L-1.

Have you submitted the raw RNA-seq data to any repository?

Line 85-105, this section is all about KEGG and GO results. The authors did not explain the results of the number of DEGs and related information. Please carefully check and add DEG-related information.

In fig 3 and 4, please define the gene name abbreviations in the captions.

Line 142-153, please properly cite Fig 5a/b/c in the text at the suitable place.

There are several spacing errors in the text.

Table 2 and 3 are not cited anywhere in the text.

The English language needs improvement.

It would have been a nice addition if the authors can add a mechanistic flow diagram showing the molecular mechanism controlling regeneration.

The revised can be considered for possible publication after minor revision.

Reviewer #3: 1. The motivation of the study is interesting, however, the hypothesis is not clearly described. In Line 263, the hypothesis is not an accurate hypothesis.

2. In the manuscript, ‘genetic transcript regulation’ of the title is not proper, because there is no temporal comparison of the transcript level or upstream/downstream relationship of gene expression to explain the transcriptional regulation.

3. There is no clear description in the Materials & Methods.

Please define the plant materials of regenerating callus and callus.

In “VOCs Collection and analysis”, the material quantity used for VOC collection and analysis and the analysis methods for quantitative data.

There is no description about RNA isolation.

Before “Sequencing data filtering and transcript assembly”, there is no description about RNA sequencing. Please describe the method clearly, including the plant materials and the gene expression comparison between RG and CL or Cyclohexane and CL.

There is no description about the application of VOC on callus, including cyclohexane, 4-methyl-2-pentanol and 1, 3-dimethyl benzene. The dosage? The quantity of callus in the treatment?...

4. The callus induction medium (2 mg/L BA) is different from the formula (1 mg/L BA) in the cited reference 35 and 36, please explain.

5. Title of Table 2, 3, 4 and Fig. 3, 4, 6 needs to be revised. For example, Table 2. Gene expression in plant regeneration of Auxin. What is “plant regeneration of auxin”? Please revised them.

6. Line 150-152, please check the exact fold change of cyclohexane.

7. Line 218, Genome?. There is no study of genome sequencing in this study.

8. There is no scale bar shown in Fig. 1a and Fig. 6.

9. Line 255 and Line 261, “the regeneration of callus” and “the root formation”. Exact morphological change of callus treated with cyclohexane is the root differentiation or induction from callus in this study. The term “the regeneration of callus” may be ambiguous with “the plant regeneration from callus”. Please define and specify the different terms in the study.

10. I sincerely suggested that the authors use a professional editing service to avoid mistakes in English scientific writing and typing error.

Line 27, lower case of G in Genetic

Line 56-58, the sentences are not complete. vitro-> in vitro; novo-> de novo; incubate->incubated; callus-> callus induction….

Line 95, wrong spelling of repaire

Line 285, formated?

Line 245, and?

6. PLOS authors have the option to publish the peer review history of their article (what does this mean?). If published, this will include your full peer review and any attached files.

Reviewer #1: No

Reviewer #2: No

Reviewer #3: No

---

## [Decision Letter · Decision Letter 1]

7 Oct 2021

PONE-D-21-17629R1Regeneration of duckweed (Lemna turonifera) involves genetic molecular regulation and cyclohexane releasePLOS ONE

Dear Dr. Sun,

Thank you for submitting your manuscript to PLOS ONE. After careful consideration, we feel that it has merit but does not fully meet PLOS ONE’s publication criteria as it currently stands. Therefore, we invite you to submit a revised version of the manuscript that addresses the points raised during the review process.

We look forward to receiving your revised manuscript.

Kind regards,

Jen-Tsung Chen, Ph.D.

Academic Editor

PLOS ONE

Journal Requirements:

Reviewers' comments:

Reviewer's Responses to Questions

**Comments to the Author**

1. If the authors have adequately addressed your comments raised in a previous round of review and you feel that this manuscript is now acceptable for publication, you may indicate that here to bypass the “Comments to the Author” section, enter your conflict of interest statement in the “Confidential to Editor” section, and submit your "Accept" recommendation.

Reviewer #1: (No Response)

Reviewer #2: All comments have been addressed

Reviewer #3: All comments have been addressed

2. Is the manuscript technically sound, and do the data support the conclusions?

Reviewer #1: No

Reviewer #2: Yes

Reviewer #3: Yes

3. Has the statistical analysis been performed appropriately and rigorously? 

Reviewer #1: No

Reviewer #2: Yes

Reviewer #3: No

4. Have the authors made all data underlying the findings in their manuscript fully available?

Reviewer #1: No

Reviewer #2: Yes

Reviewer #3: Yes

5. Is the manuscript presented in an intelligible fashion and written in standard English?

Reviewer #1: No

Reviewer #2: Yes

Reviewer #3: Yes

7. PLOS authors have the option to publish the peer review history of their article (what does this mean?). If published, this will include your full peer review and any attached files.

Reviewer #1: No

Reviewer #2: No

Reviewer #3: No

6. Review Comments to the Author

Reviewer #2: The authors have satisfactorily addressed all the comments, and I have no more comments. Thus, I endorsed the submission for acceptance in its current form.

Congratulations!

Reviewer #3: The fulltext has been carefully revised according to the suggestions and comments of the reviewers and improved the quality of the manuscript. In my opinion, the manuscript can be acceptable for publication but after the minor revision as suggested.

1. For the description of fold change in RNA sequencing data, the 4nd decimal place seems redundant. Please consider to round off to second or third decimal place seems redundant.

2. Line 142, please substitute symbol “x” to * in the fulltext.

3. Please improve the format of Table 2, 3 and 4.

---

## [Author Response · Author response to Decision Letter 1]

7 Oct 2021

Dear editor：

Thank you very much for your kindly help in processing the review of our manuscript. We have carefully read these thoughtful comments from you and reviews, and the minor revision has been done. We appreciate that our research and revision has been affirmed by reviewers. Reviewer 2 said that the authors have satisfactorily addressed all the comments, and I have no more comments. Thus, I endorsed the submission for acceptance in its current form. Reviewer 3 said the manuscript can be acceptable for publication after the minor revision as suggested. Thanks so much for the valuable advice and help from reviewers. We consider all issues mentioned in the reviewers' comments carefully, and made a minor revision as following. 

1.For the description of fold change in RNA sequencing data, the 4nd decimal place seems redundant. Please consider to round off to second or third decimal place seems redundant.

Reply: For the description of fold change, the 4nd decimal place has been round off to second decimal place：

Line 107: In this study, AUX/IAA and ARF have been significantly down-regulated, by 13.03 and 3.01 log2 fold change, respectively. 

Line 115: The expression of ZFP was decreased by 4.04 log2 fold change.

Line 120: Histidine phosphate transfer protein (AHP), which interacts with CRE1 and CKI1, was up-regulated by 2.97 log2 fold change.

Line 124: A-ARR was down-regulated by 4.53 log2 fold change, which might lead to overall up-regulation in cytokinins during callus regeneration.

2.Line 142, please substitute symbol “x” to * in the fulltext.

Reply: we have substituted symbol “x” to *: The peak area of 1, 3-dimethyl benzene in the regenerating callus was 0.84*107

3.Please improve the format of Table 2, 3 and 4.

Reply: The format of Table 2, 3 and 4 have been improved.

---

## [Editor Report · Decision Letter 2]

12 Oct 2021

Regeneration of duckweed (Lemna turonifera) involves genetic molecular regulation and cyclohexane release

PONE-D-21-17629R2

Dear Dr. Sun,

We’re pleased to inform you that your manuscript has been judged scientifically suitable for publication and will be formally accepted for publication once it meets all outstanding technical requirements.

Kind regards,

Jen-Tsung Chen, Ph.D.

Academic Editor

PLOS ONE
---

## [Editor Report · Acceptance letter]

27 Dec 2021

PONE-D-21-17629R2 

Regeneration of duckweed *(Lemna turonifera)* involves genetic molecular regulation  and cyclohexane release 

Dear Dr. Sun:

I'm pleased to inform you that your manuscript has been deemed suitable for publication in PLOS ONE. Congratulations! Your manuscript is now with our production department. 

Kind regards, 

on behalf of

Dr. Jen-Tsung Chen 

Academic Editor

PLOS ONE